# Native Low-Density Lipoproteins Act in Synergy with Lipopolysaccharide to Alter the Balance of Human Monocyte Subsets and Their Ability to Produce IL-1 Beta, CCR2, and CX3CR1 In Vitro and In Vivo: Implications in Atherogenesis

**DOI:** 10.3390/biom11081169

**Published:** 2021-08-07

**Authors:** Aarón N. Manjarrez-Reyna, Camilo P. Martínez-Reyes, José A. Aguayo-Guerrero, Lucia A. Méndez-García, Marcela Esquivel-Velázquez, Sonia León-Cabrera, Gilberto Vargas-Alarcón, José M. Fragoso, Elizabeth Carreón-Torres, Oscar Pérez-Méndez, Jessica L. Prieto-Chávez, Galileo Escobedo

**Affiliations:** 1Laboratory of Immunometabolism, Research Division, General Hospital of Mexico “Dr. Eduardo Liceaga”, Mexico City 06720, Mexico; aaron.manjarrez@gmail.com (A.N.M.-R.); nava111222@hotmail.com (C.P.M.-R.); jose.aguayo01@iest.edu.mx (J.A.A.-G.); angelica.mendez.86@hotmail.com (L.A.M.-G.); esquivel.marcela@gmail.com (M.E.-V.); 2Unidad de Biomedicina, Facultad de Estudios Superiores-Iztacala, Universidad Nacional Autónoma de México, Tlalnepantla, Edo. De Mexico 54090, Mexico; soleon81@gmail.com; 3Carrera de Médico Cirujano, Facultad de Estudios Superiores-Iztacala, Universidad Nacional Autónoma de México, Tlalnepantla, Edo. De Mexico 54090, Mexico; 4Department of Molecular Biology, Instituto Nacional de Cardiología “Ignacio Chávez”, Juan Badiano 1, Sección XVI, Tlalpan, Mexico City 14080, Mexico; gvargas63@yahoo.com (G.V.-A.); mfragoso1275@yahoo.com.mx (J.M.F.); qfbelizabethcm@yahoo.es (E.C.-T.); opmendez@yahoo.com (O.P.-M.); 5School of Engineering and Sciences Campus CDMX, Tecnológico de Monterrey, Calle Puente 222, Tlalpan, Mexico City 14380, Mexico; 6Laboratorio de Citometría de Flujo, Centro de Instrumentos, Coordinación de Investigación en Salud, Hospital de Especialidades del Centro Médico Siglo XXI, Instituto Mexicano del Seguro Social, Mexico City 06720, Mexico; lakshmi.litmus@hotmail.com

**Keywords:** atherogenesis, monocyte subpopulations, native LDL, LPS, IL-1 beta, CCR2, CX3CR1, LBP

## Abstract

Increasing evidence has demonstrated that oxidized low-density lipoproteins (oxLDL) and lipopolysaccharide (LPS) enhance accumulation of interleukin (IL)-1 beta-producing macrophages in atherosclerotic lesions. However, the potential synergistic effect of native LDL (nLDL) and LPS on the inflammatory ability and migration pattern of monocyte subpopulations remains elusive and is examined here. In vitro, whole blood cells from healthy donors (n = 20) were incubated with 100 μg/mL nLDL, 10 ng/mL LPS, or nLDL + LPS for 9 h. Flow cytometry assays revealed that nLDL significantly decreases the classical monocyte (CM) percentage and increases the non-classical monocyte (NCM) subset. While nLDL + LPS significantly increased the number of NCMs expressing IL-1 beta and the C-C chemokine receptor type 2 (CCR2), the amount of NCMs expressing the CX3C chemokine receptor 1 (CX3CR1) decreased. In vivo, patients (n = 85) with serum LDL-cholesterol (LDL-C) >100 mg/dL showed an increase in NCM, IL-1 beta, LPS-binding protein (LBP), and Castelli’s atherogenic risk index as compared to controls (n = 65) with optimal LDL-C concentrations (≤100 mg/dL). This work demonstrates for the first time that nLDL acts in synergy with LPS to alter the balance of human monocyte subsets and their ability to produce inflammatory cytokines and chemokine receptors with prominent roles in atherogenesis.

## 1. Introduction

Atherosclerosis is a chronic inflammatory disease of the arteries and a leading cause of death worldwide [1]. Atherogenesis is the pathological process through which atheromatous plaque is formed in the inner layer of the arteries, leading to vessel thickening, arterial remodeling, and potential obstruction of blood flow at the site of the lesion [2].

Atherosclerotic plaque formation is a highly dynamic process involving the adhesion of circulating monocytes to the tunica intima, wherein these cells differentiate into macrophages [3]. Then, monocyte-derived macrophages can migrate to the subendothelial space and turn into foam cells after ingesting oxidized low-density lipoproteins (oxLDL), triggering an inflammatory response that injures endothelial cells and promotes arterial remodeling [4]. In this scenario, the role of macrophages and oxLDL in atherogenesis appears to be clear [5]; however, monocytes and native LDL (nLDL) might also contribute to atherosclerosis by mechanisms that are not yet fully understood.

In the blood stream, human monocytes are sorted into three subsets based on the cell surface expression of CD14 and CD16 [6]. Classical monocytes (CMs) express high CD14 levels but do not express CD16 (CD14++CD16−). Intermediate monocytes (IMs) exhibit CD16 expression and high CD14 levels (CD14++CD16+), while non-classical monocytes (NCMs) also express CD16 but show low CD14 levels (CD14+CD16+) [7]. Upon lipopolysaccharide (LPS) stimulation, human monocyte subpopulations differentially respond to produce interleukin (IL)-1 beta, a proinflammatory cytokine with key roles in atherosclerosis [8]. Monocyte subsets also differentially express the C-C chemokine receptor type 2 (CCR2) and the CX3C chemokine receptor 1 (CX3CR1), each of which are chemokine receptors with prominent functions in cell migration and endothelial adhesion during atherosclerosis [9,10].

The immune function of monocyte subpopulations is regulated by prototypical factors such as LPS, double-stranded ribonucleic acid (dsRNA), and tumor necrosis factor alpha (TNF-alpha) [11,12]. However, emerging evidence suggests that non-prototypical immunometabolic ligands can also influence the cytokine and chemokine expression profile in these cells. In this sense, excess glucose increases TNF-alpha expression in in vitro cultured primary human monocytes [13]. Free-fatty acids induce IL-1 beta secretion in in vitro cultured THP-1 cells and primary human monocytes [14]. Furthermore, immunometabolic agents can also act in synergy with prototypical immune factors to regulate the activity of monocyte subsets. In this regard, low concentration of high-density lipoproteins (HDL) increases IL-1 beta expression in NCMs stimulated with LPS and in subjects with high serum levels of LPS-binding protein (LBP) [15,16,17]. On the contrary, the study of the action of LDL in monocytes has been restricted to its role as oxLDL [4,5], even though accumulating evidence also suggests that monocytes and nLDL could interact in circulation, thus contributing to atherosclerotic plaque formation [18,19]. In this sense, elevated circulating nLDL levels are associated with an increment in the percentage of CMs that can migrate into endothelial tissue by mainly expressing CX3CR1 and CCR2 in *ApoE^−/−^* mice, an animal model of atherosclerosis [20]. In parallel, LPS directly increases lipid deposition in primary human adventitial fibroblasts, inducing secretion of molecules with prominent roles in atherosclerosis such as monocyte chemoattractant protein 1 (MCP-1), the main ligand for CCR2 [21]. Interestingly, consumption of Western-style high-fat diets is associated with increased serum LPS levels in humans [22,23]. However, the potential contribution of nLDL and LPS to the inflammatory activity of human monocyte subsets during atherogenesis remains unclear.

The main goal of this study was to examine the effect of nLDL on the immune function of human monocyte subpopulations in in vitro LPS-stimulated primary monocytes and in patients with high LDL-cholesterol (LDL-C) and LBP serum levels.

## 2. Materials and Methods

### 2.1. In Vitro Culture of Primary Human Monocytes

Twenty healthy blood donors with LDL-C serum levels less than 100 mg/dL and high-sensitive C-reactive protein (hs-CRP) serum values of 1.35 ± 0.26 mg/L, on average, were enrolled in the study. Each participant agreed to donate 8 mL of blood, which was collected into a tube containing heparin (Vacutainer^TM^, BD Diagnostics, Franklin Lakes, NJ, USA). Subsequently, whole blood samples were individually divided and placed in 6-well cell-culture plates (Costar, Kennebunk, ME, USA), adding 2 mL of blood plus 1 mL of RPMI-1640 (Sigma-Aldrich, St. Louis, MO, USA) supplemented with 5% fetal bovine serum (FBS), 2 mM L-glutamine, 10 nM HEPES buffer, and 50 μg/mL gentamicin (Gibco^TM^, Grand Island, NY, USA) per well. The blood sample contained in the first well was designated as the control and received 300 μL of RPMI-1640 for 9 h. The second well was incubated in the presence of 100 μg nLDL (Sigma-Aldrich, St. Louis, MO, USA) dissolved in 300 μL of RPMI-1640 for 9 h. The third well was incubated in the presence of 10 ng/mL LPS (Sigma-Aldrich, St. Louis, MO, USA) dissolved in 300 μL of RPMI-1640 for 9 h. The sample contained in the fourth well was incubated in the presence of 100 μg nLDL plus 10 ng/mL LPS dissolved in 300 μL of RPMI-1640 for 9 h. Exposure time of in vitro cultures was selected based on time-response curves at 3, 6, and 9 h, finding that monocytes show the most significant changes at 9 h. The cell-culture plates were incubated at 37 °C in humidified 5% CO_2_ atmosphere. For intracellular cytokine stain, white blood cells (WBCs) were treated with 1:1000 Brefeldin A (BioLegend, Inc., San Diego, CA, USA) for the last 2 h of in vitro culture. All of the participants provided written informed consent, previously approved by the institutional ethical committee of the General Hospital of Mexico (registration number of ethical approval code: DI/20/501/03/17), which guaranteed that the study was conducted in rigorous adherence to the principles described in the 1964 Declaration of Helsinki and its posterior amendment in 2013.

### 2.2. Flow Cytometry

After incubation, whole blood samples were collected and centrifuged at 500× *g* for 10 min. Immediately afterwards, WBCs were separated using a micropipette and resuspended in 1 mL PBS1X (Sigma-Aldrich, St. Louis, MO, USA). After an additional centrifugation step and removal of the supernatant, each cell pellet was resuspended in 50 μL cell staining buffer (BioLegend, Inc., San Diego, CA, USA). WBCs were incubated with 5 μL True-Stain Monocyte Blocker^TM^ (BioLegend, Inc., San Diego, CA, USA) for 10 min on ice. Then, WBCs were incubated with anti-CD14 PE/Cy7, anti-CD16 PE/Cy5, anti-CCR2 AF647, anti-CX3CR1 BV510, Zombie ultraviolet (UV) Fixable Viability Kit (BioLegend, Inc., San Diego, CA, USA), and anti-human leukocyte antigen-DR (HLA-DR) BUV661 (BD Biosciences, San Jose, CA, USA) for 20 min in darkness at 4 °C. Afterwards, WBCs were incubated with 100 μL Fixation Medium A (FIX & PERM^TM^ Cell Permeabilization Kit) (Invitrogen^TM^, Carlsbad, CA, USA) for 20 min at room temperature. After being rinsed using Cell Staining Buffer (BioLegend, Inc., San Diego, CA, USA), peripheral blood mononuclear cells (PBMCs) were incubated with 100 μL Permeabilization Medium B (FIX & PERM^TM^ Cell Permeabilization Kit, Invitrogen^TM^, Carlsbad, CA, USA) and anti-IL-1 beta Pacific Blue (BioLegend, Inc., San Diego, CA, USA) for 20 min in darkness at room temperature. After being rinsed using Cell Staining Buffer (BioLegend, Inc., San Diego, CA, USA), PBMCs were acquired on a BD Influx flow cytometer (BD Biosciences, San Jose, CA, USA) using the BD Sortware^TM^ software 1.2, acquiring 20,000 events per test in triplicate. For the gating strategy, WBCs were first gated on a time/side scatter density plot, and then gated on the Zombie UV negative cell population for detection of living cells. Afterwards, living cells were gated for singlets on a forward scatter (FS)/Trigger Pulse Width density plot. Monocytes were recognized on the HLA-DR gating. Then, monocytes were selected using the rectangular gating strategy on the CD14+/CD16+ population for identification of CMs (CD14++CD16−), IMs (CD14++CD16+), and NCMs (CD14+CD16+), as previously reported [24]. The median fluorescence intensity (MFI) for IL-1 beta, CCR2, and CX3CR1 was obtained by considering both positive and negative cell populations for each marker. The percentage of positive cells for each marker was obtained using proper fluorescence minus one (FMO) controls. Compensation controls were performed by means of UltraComp eBeads^TM^ (Invitrogen^TM^, Carlsbad, CA, USA) for each fluorochrome. Data were analyzed by means of the FlowJo 10.0.7 software (TreeStar, Inc., Ashland, OR, USA).

### 2.3. Subjects for In Vivo Assays

One hundred fifty volunteers from both sexes, aged 18 years or older, with 8 h fasting who attended the Blood Bank and the Department of Internal Medicine of the General Hospital of Mexico were included in the study. A group of three trained physicians registered gender, age, body mass index (BMI), waist circumference, body fat percentage, and the serum levels of glucose, insulin, C-reactive protein (CRP), total cholesterol, triglycerides, HDL-C, and LDL-C in all participants. BMI resulted from dividing weight by height squared. Waist circumference was measured at the midpoint between the lower rib margin and the iliac crest using a tape. Body fat percentage was obtained by means of a body composition analyzer (TANITA^®^ Body Composition Analyzer, Model TBF-300A, Tokyo, Japan). CRP was measured in triplicate by immunoturbidimetry (Randox Laboratories, Meenmore, Ireland). Serum glucose, total cholesterol, triglycerides, HDL-C, and LDL-C were measured in triplicate by enzymatic assays (Roche Diagnostics, Mannheim, Germany). Serum insulin was measured in triplicate by the Enzyme-Linked Immunosorbent Assay (ELISA) (Abnova, Corporation, Taipei City, Taiwan). The estimate of insulin resistance was individually calculated using the homeostatic model assessment of insulin resistance (HOMA-IR) by multiplying glucose concentration (mM) by insulin concentration (mU/L) and then dividing by 22.5. All of the participants received full medical evaluation and provided written informed consent, previously approved by the institutional ethical committee of the General Hospital of Mexico (registration number of ethical approval code: DIC/11/UME/05/029), which guaranteed that the study was conducted in rigorous adherence to the principles described in the 1964 Declaration of Helsinki and its posterior amendment in 2013. Volunteers were excluded from the study if they had a previous diagnosis of type 1 diabetes (T1D), T2D, coronary disease, acute or chronic liver or renal disease, cancer, endocrine disorders, infectious diseases, chronic inflammatory disease, and/or autoimmune disorders. We also excluded from the study human immunodeficiency virus (HIV), hepatitis C virus (HCV), and hepatitis B virus (HBV)-seropositive patients, pregnant or lactating woman, and subjects with anti-inflammatory or immunomodulatory medication, including non-steroidal anti-inflammatory drugs (NSAIDs).

### 2.4. Effects of LDL and LBP on Monocyte Subpopulations and IL-1 Beta In Vivo

According to clinical guidelines of the National Cholesterol Education Program (NCEP) Expert Panel on Detection, Evaluation and Treatment of High Blood Cholesterol in Adults (Adult Treatment Panel III), the study participants were divided in two groups of LDL-C concentration, as follows: subjects with optimal concentration of LDL-C ≤ 100 mg/dL, and individuals with high concentration of LDL-C > 100 mg/dL [25]. Then, BMI, waist circumference, body fat percentage, glucose metabolism, HOMA-IR, and lipid profile were registered and compared between both groups. Blood samples (6 mL) were obtained from all volunteers for posterior isolation of WBCs. WBCs were incubated with anti-CD14 PE/Cy7 and anti-CD16 FITC, acquiring 20,000 events per test in triplicate on a BD Influx flow cytometer (BD Biosciences, San Jose, CA, USA) using the BD Sortware^TM^ software 1.2 and the FlowJo 10.0.7 software (TreeStar, Inc., Ashland, OR, USA), as described above. In all participants, serum IL-1 beta (Peprotech, Mexico City, Mexico) and LBP (Invitrogen^TM^, Carlsbad, CA, USA) were measured in triplicate by ELISA and analyzed according to LDL-C levels.

### 2.5. Statistics

Normality of data distribution was estimated by the Shapiro–Wilk test. For in vitro assays, one-way ANOVA, followed by a post-hoc Tukey test, was used to compare percentages of CMs, IMs, and NCMs, and expression of IL-1 beta, CCR2, and CX3CR1 in the cell groups designated as control, nLDL, LPS, and nLDL + LPS. For in vivo assays, the unpaired Student’s *t*-test was used to compare subjects with serum LDL-C ≤ 100 mg/dL and individuals with serum LDL-C > 100 mg/dL in terms of gender, age, BMI, waist circumference, body fat percentage, fasting glucose, insulin, HOMA-IR, CRP, total cholesterol, triglycerides, HDL, percentages of CMs, IMs, and NCMs, serum IL-1 beta, and circulating LBP levels. Differences were considered significant when *p* < 0.05. All of the statistical analyses were performed by means of the GraphPad Prism 7 software (GraphPad Software, La Jolla, CA 92037, USA).

## 3. Results

For the gating strategy, WBCs were first gated on a time/side scatter (SS) density plot, and then gated on the Zombie UV negative cell population for detection of living WBCs (Figure 1, top panel). Afterwards, WBCs were gated for singlets on a forward scatter (FS)/Trigger Pulse Width density plot, and after that gated on the HLA-DR+ population for monocyte recognition. Monocytes were then gated on the CD14+/CD16+ population for identification of CMs (CD14++CD16−), IMs (CD14++CD16+), and NCMs (CD14+CD16+). Monocyte subsets were then gated on the IL-1 beta+, CCR2+, and CX3CR1+ populations for assessing the effects of nLDL and LPS on the immune activity of these cells (Figure 1, bottom panel).

Figure 2 illustrates representative plots showing the percentages of CMs, IMs, and NCMs treated with nLDL and/or LPS (Figure 2A–D). As compared to control cells, the percentage of CMs was significantly reduced when treated with 100 μg/mL nLDL (Figure 2E). While the percentage of IMs did not change (Figure 2F), the percentage of NCMs exhibited a significant 20% increase when exposed to 100 μg/mL nLDL with respect to that found in untreated cells (Figure 2G). Monocyte subpopulations also differentially responded to LPS. LPS, alone or in combination with nLDL, induced a significant 10% increase in CMs (Figure 2E). Conversely, IMs and NCMs showed significant reductions when treated with LPS or nLDL + LPS with respect to that found in control cells (Figure 2F,G, respectively). The percentages of CMs, IMs, and NCMs showed similar behavior in response to LPS or nLDL + LPS, suggesting that this effect was mainly mediated by LPS (Figure 2E–G, respectively).

Exposure of cells to nLDL and LPS not only affected the balance among monocyte subpopulations but also their ability to produce IL-1 beta, a cytokine with key proinflammatory actions. Exposure of monocytes to 100 μg/mL nLDL had no significant effects on IL-1 beta production in all monocyte subpopulations (Figure 3A). LPS significantly increased IL-1 beta production in CMs, IMs, and NCMs, and this increase was even more evident when cells were exposed to LPS in combination with nLDL (Figure 3A). In parallel, the percentages of CMs, IMs, and NCMs that expressed IL-1 beta also tended to decrease in response to nLDL (Figure 3B). LPS, alone or in combination with nLDL, significantly increased the number of IL-1 beta+ cells in all monocyte subpopulations with respect to untreated cells (Figure 3B).

Besides IL-1 beta expression, monocyte subsets have also been shown to differentially express chemokine receptors. In this sense, CCR2 expression was clearly higher in CMs than IMs and NCMs; however, CMs did not differentially express CCR2 in response to nLDL, alone or in combination with LPS (Figure 4A). Overall, the percentage of CMs and IMs expressing CCR2 did not change in response to nLDL or LPS (Figure 4B, left and middle panels, respectively). Conversely, nLDL + LPS significantly increased the amount of NCMs expressing CCR2 as compared to that found when cells were only exposed to nLDL (Figure 4B, right panel). Expression of CX3CR1 was significantly higher in IMs and NCMs than CMs (Figure 4C). Furthermore, nLDL induced a decrease in CX3CR1 expression that was even more evident in NCMs than CMs and IMs (Figure 4C, right panel). The exposure of cells to LPS, alone or in combination with nLDL, decreased the expression of CX3CR1 in all monocyte subpopulations (Figure 4C). Interestingly, the number of CX3CR1+ NCMs significantly decreased in response to nLDL (Figure 4D, right panel), while LPS together with nLDL had the same effect upon IMs and NCMs (Figure 4D, middle and right panels, respectively).

In vitro, nLDL and LPS not only changed the balance of monocyte subsets but also the expression pattern of molecules that have been previously associated with the inflammatory response in atherogenesis. Thus, we decided to confirm these results by conducting similar experiments but now with an in vivo approach in patients with elevated LDL-C serum levels. Table 1 summarizes the differences in metabolic syndrome-related risk factors between subjects with normal and elevated LDL-C serum levels (76.53 ± 4.1 vs. 128.7 ± 3.6, respectively). There were no significant differences between subjects with LDL-C values below and above 100 mg/dL with respect to sex proportion, age, BMI, waist circumference, body fat percentage, triglycerides, HDL-C, fasting insulin and glucose, HOMA-IR, and CRP (Table 1). On the contrary, total cholesterol and Castelli’s cardiovascular risk index II significantly increased in subjects with LDL-C greater than 100 mg/dL (Table 1).

Confirming our in vitro results, the CM percentage significantly decreased in subjects with LDL-C > 100 mg/dL (Figure 5A). Although the IM percentage showed no differences (Figure 5B), the NCM percentage exhibited a significant 1.5-fold increase in individuals with LDL-C greater than 100 mg/dL (Figure 5C). The serum levels of IL-1 beta also significantly increased in subjects with LDL-C values above 100 mg/dL (Figure 5D). Moreover, LBP levels clearly raised in subjects with LDL-C > 100 mg/dL, suggesting the presence of abnormally high LPS serum values in these individuals (Figure 5E).

## 4. Discussion

Macrophages and oxLDL are considered to be the main contributors to foam cell formation and inflammatory cell infiltration in atherosclerotic lesions [18]. However, the fact that monocytes and nLDL are able to interact in the blood stream has raised the question as to whether these precursors might also play a pivotal role in atherogenesis [19]. In this sense, intraperitoneal administration of triglyceride-rich lipoproteins decreases the number of non-classical Ly6C/Gr1low monocytes in mice [26]. Similarly, dyslipidemia with elevated levels of either LDL or very low-density lipoproteins (VLDL) increases both classical Ly6C/Gr1high and non-classical Ly6C/Gr1low monocyte subsets in mice fed a high-fat diet [19]. In line with this evidence, our results demonstrate that nLDL can alter the balance of human monocyte subpopulations by directly decreasing CMs and increasing NCMs in vitro and in vivo.

The mechanism through which nLDL induce imbalance of monocyte subsets remains unclear, and it probably involves CD14 and CD16 expression. As mentioned, CD14 and CD16 expression is the primary feature that defines monocyte subpopulations in humans [27]. CD14 is a transmembrane protein that forms the LPS receptor complex together with Toll-like receptor 4 (TLR4) and the myeloid differentiation factor 2 (MD-2) [28]. The trimeric CD14/TLR4/MD-2 complex triggers the nuclear factor kappa B (NFκB)-dependent signaling pathway in charge of regulating the expression of inflammatory cytokines such as IL-6, TNF-alpha, and IL-1 beta [29]. CD14 expression is regulated by specificity protein 1 (Sp1), a transcription factor that is activated via LDL receptor (LDLr) promoter [30,31]. This body of evidence supports the idea that nLDL could decrease the CM proportion and increase the NCM percentage by regulating CD14 expression, probably via LDLr-Sp1. CD16 is an Fc-gamma receptor that recognizes IgG-coated microorganisms and immune complexes [32]. Recent evidence has demonstrated that CD16 expression is under the post-transcriptional control of microRNA (miR)-218 in human natural killer (NK) cells [33]. In vitro, oxLDL are able to decrease miR-218 expression in cardiac microvascular endothelial cells from rats. In vivo, miR-218 expression is inversely correlated with oxLDL levels in patients with coronary artery disease (CAD) [34]. In this sense, the increase in CD14+CD16+ non-classical monocytes and the decrease in CD14+CD16− classical monocytes that we found may be potentially related to the miR-218 suppression that in turn favors CD16 expression, promoting conversion of classical monocytes into non-classical monocytes in response to nLDL. Thus, circulating nLDL may induce imbalance of human monocyte subpopulations by mechanisms able to regulate CD14 and CD16 expression such as those orchestrated by LDLr, Sp1, and miR-218; however, this hypothesis remains to be elucidated in further studies.

A solid body of evidence has demonstrated that human monocyte subsets display important inflammatory actions by preferably expressing IL-1 beta in response to LPS [8,35]. LPS is a component of the outer membrane of Gram-negative bacteria that can translocate across the intestinal wall to the hepatic portal circulation, wherein it binds to LBP [36]. The LPS/LBP complex is considered a key inflammatory trigger for monocytes and macrophages and has been found to be increased in obese subjects with atherosclerotic heart disease, among whom nLDL is also increased [37,38]. Upon recognition, LPS induces the NFκB-dependent synthesis of pro-IL-1 beta, which in turn is proteolytically excised by the NOD-, LRR-, and pyrin domain-containing protein 3 (NLRP3) inflammasome to form mature IL-1 beta [39]. IL-1 beta has multiple functions in atherogenesis. In human microvascular endothelial cells, IL-1 beta stimulates the expression of the intracellular adhesion molecule-1 (ICAM-1) and the vascular cell adhesion molecule-1 (VCAM-1), which have the ability to recruit leukocytes into the inner layer of arteries [40]. In human aorta smooth muscle cells, IL-1 beta induces the expression of the monocyte chemoattractant protein-1 (MCP-1), which in turn contributes to mononuclear cell migration toward the vascular endothelium [41]. IL-1 beta also directly promotes atherosclerotic lesion formation by stimulating production of platelet-derived growth factor, a molecule with the ability to induce proliferation of vascular smooth muscle cells in the atheroma [42]. Interestingly, the exposure of endothelial cells and vascular smooth muscle cells to LPS increases release of IL-1 beta in atherosclerotic plaques of patients with suboptimally controlled hyperlipidemia LDL [43]. Our data expand on this body of evidence by demonstrating for the first time that LPS acts in synergy with nLDL to increase IL-1 beta synthesis in in vitro cultured human monocytes, especially CMs and IMs. In vivo, our results confirm this finding by revealing that subjects with elevated LDL-C serum levels also display higher LBP, IL-1 beta, and atherogenic index than those found in individuals with optimal LDL-C concentrations. However, it is worth mentioning that we still need to assess the potential effect of LDL particle number and size on the dynamics of human monocyte subpopulations and their ability to produce IL-1 beta, which may expand on the knowledge regarding the multiple actions of LDL in the inflammatory response associated with atherosclerosis.

nLDL and LPS not only displayed synergistic effects on the distribution of human monocyte subsets and their ability to produce IL-1 beta, but also affected the expression pattern of chemokine receptors such as CCR2 and CX3CR1. CCR2 is a C-C chemokine receptor mainly expressed in mononuclear cells that mediates migration of monocytes to inflamed tissues in response to MCP-1 [44]. CX3CR1 is the cell receptor for fractalkine, a chemokine with the ability to induce retention of monocytes in circulation, preferably those expressing CD16 such as IMs and NCMs [45]. In our study, the number of CCR2+ NCMs increased in response to nLDL and LPS, which may act in favor of promoting recruitment of these cells to the vascular endothelium. In line with these findings, a seminal work conducted by Han et al. informed that CCR2 expression increases in primary monocytes from patients with elevated LDL-C serum values with respect to normocholesterolemic controls [46]. Additionally, exposure of THP-1 monocytes to nLDL induces a significant increase in both CCR2 expression and in vitro chemotactic response to MCP-1, suggesting that nLDL can directly enhance monocyte recruitment to the atherosclerotic lesion [46]. Concurring with the idea of increased cell recruitment, our data show that nLDL and LPS also decreased the amount of CX3CR1+ NCMs. In this sense, Nielsen et al. found altered expression of TNF-alpha and CX3CR1 in primary monocytes from patients with familial hypercholesterolemia as compared to subjects with normal LDL-C levels [47]. Interestingly, TNF-alpha and CX3CR1 expression positively correlated with increased intima-media thickness and hs-CRP, both of which are markers of atherosclerosis and cardiovascular risk [47]. In parallel, Geng et al. demonstrated that LPS administration to *ApoE^−/−^* mice enhances monocyte recruitment and macrophage accumulation in aortic atherosclerotic plaques, which concurs with increased lipid deposition in the atheroma [48]. Altogether, these findings support the idea that nLDL and LPS may directly contribute to atherogenesis by altering the expression pattern of inflammatory cytokines and chemokine receptors in monocytes, which may promote recruitment of these cells to atherosclerotic plaques. However, this notion should be still tested in in vivo experimental approaches focused on characterizing how nLDL and LPS affect the recruiting pattern of IL-1 beta-, CCR2-, and CX3CR1-producing monocytes to atherosclerotic lesions.

As we have outlined here, nLDL appear to have additional proinflammatory roles in atherogenesis that differed from those classically described for oxLDL [49]. Moreover, the effects of nLDL on the inflammatory ability of human monocyte subpopulations are enhanced by LPS, confirming previous evidence suggesting that nLDL and LBP form a negative loop that contributes to atherosclerosis [50,51,52].

## 5. Conclusions

In conclusion, using in vitro and in vivo complementary assays, this work demonstrates for the first time that nLDL act in synergy with LPS to alter the balance of human monocyte subsets and their ability to produce inflammatory cytokines and chemokine receptors with prominent roles in atherogenesis.

## Figures and Tables

**Figure 1 biomolecules-11-01169-f001:**
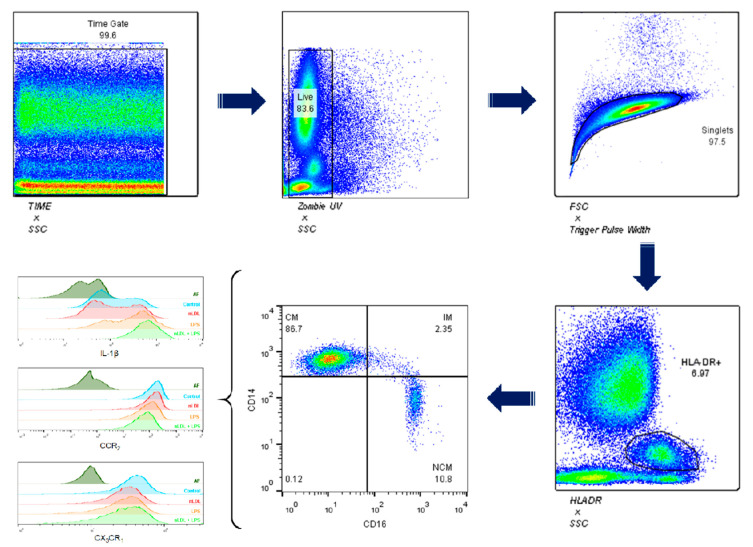
Gating strategy for characterizing human monocyte subsets. White blood cells were first gated on a time/side scatter (SS) density plot, and then gated on the Zombie UV negative cell population for detection of living cells. Afterwards, living cells were gated for singlets on a forward scatter (FS)/Trigger Pulse Width density plot. Monocytes were recognized on the HLA-DR gating. Then, monocytes were gated on the CD14+/CD16+ population for identification of CMs (CD14++CD16−), IMs (CD14++CD16+), and NCMs (CD14+CD16+). Expression of IL-1 beta, CCR2, and CX3CR1 was measured in all monocyte subsets. SSC, side scatter; FSC, forward scatter; HLA-DR, human leukocyte antigen-DR isotype; CM, classical monocytes; IM, intermediate monocytes; NCM, non-classical monocytes; IL-1 beta, interleukin 1 beta; CCR2, C-C chemokine receptor type 2; CX3CR1, CX3C chemokine receptor 1.

**Figure 2 biomolecules-11-01169-f002:**
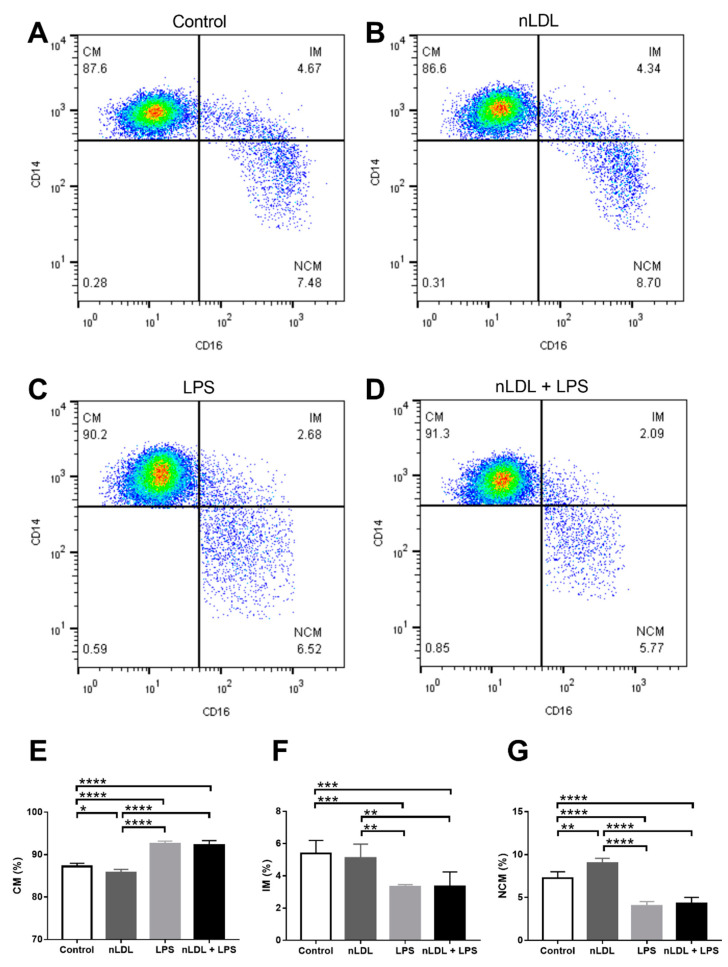
Effects of nLDL and LPS on the dynamics of human monocyte subpopulations. Representative plots showing the percentages of CMs, IMs, and NCMs in response to control conditions (**A**), nLDL (**B**), LPS (**C**), or a combination of nLDL + LPS (**D**). nLDL significantly decreased the CM percentage and increased the amount of NCMs as compared to control cells (**E**,**G**, respectively). LPS, alone or in combination with nLDL, significantly increased the CM percentage and decreased the amount of IMs and NCMs (**E**–**G**, respectively). Cells were incubated in the presence or absence of 100 μg/mL nLDL and/or 10 ng/mL LPS for 9 h. Data are expressed as mean ± standard deviation. Data were compared using one-way ANOVA followed by the post-hoc Tukey test. Differences were considered significant when *p* < 0.05. * = *p* < 0.05; ** = *p* < 0.01; *** = *p* < 0.001; **** = *p* < 0.0001. CM, classical monocytes; IM, intermediate monocytes; NCM, non-classical monocytes; nLDL, native low-density lipoproteins; LPS, lipopolysaccharide.

**Figure 3 biomolecules-11-01169-f003:**
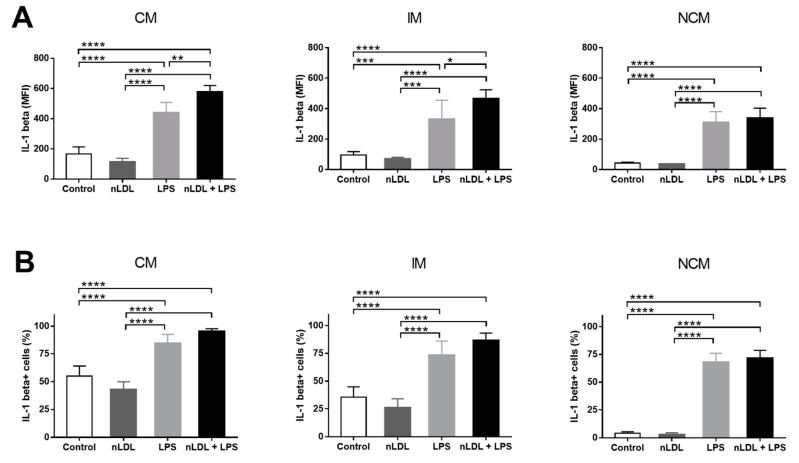
Effects of nLDL and LPS on IL-1 beta production in human monocyte subpopulations. Although no significant differences were found, nLDL tended to decrease IL-1 beta production in CMs, IMs, and NCMs (**A**, left, middle, and right panels, respectively). LPS, alone or in combination with nLDL, significantly increased IL-1 beta expression in CMs, IMs, and NCMs (**A**, left, middle, and right panels, respectively). Although no significant differences were found, nLDL tended to decrease the percentages of IL-1 beta+ cells in all monocyte subsets (**B**, left, middle, and right panels, respectively). LPS, alone or in combination with nLDL, significantly increased the amount of IL-1 beta+ cells in all monocyte subpopulations (**B**, left, middle, and right panels, respectively). Cells were incubated in the presence or absence of 100 μg/mL nLDL and/or 10 ng/mL LPS for 9 h. Data are expressed as mean ± standard deviation. Data were compared using one-way ANOVA followed by the post-hoc Tukey test. Differences were considered significant when *p* < 0.05. * = *p* < 0.05; ** = *p* < 0.01; *** = *p* < 0.001; **** = *p* < 0.0001. CM, classical monocytes; IM, intermediate monocytes; NCM, non-classical monocytes; nLDL, native low-density lipoproteins; LPS, lipopolysaccharide; IL-1 beta, interleukin 1 beta; MFI, median fluorescence intensity.

**Figure 4 biomolecules-11-01169-f004:**
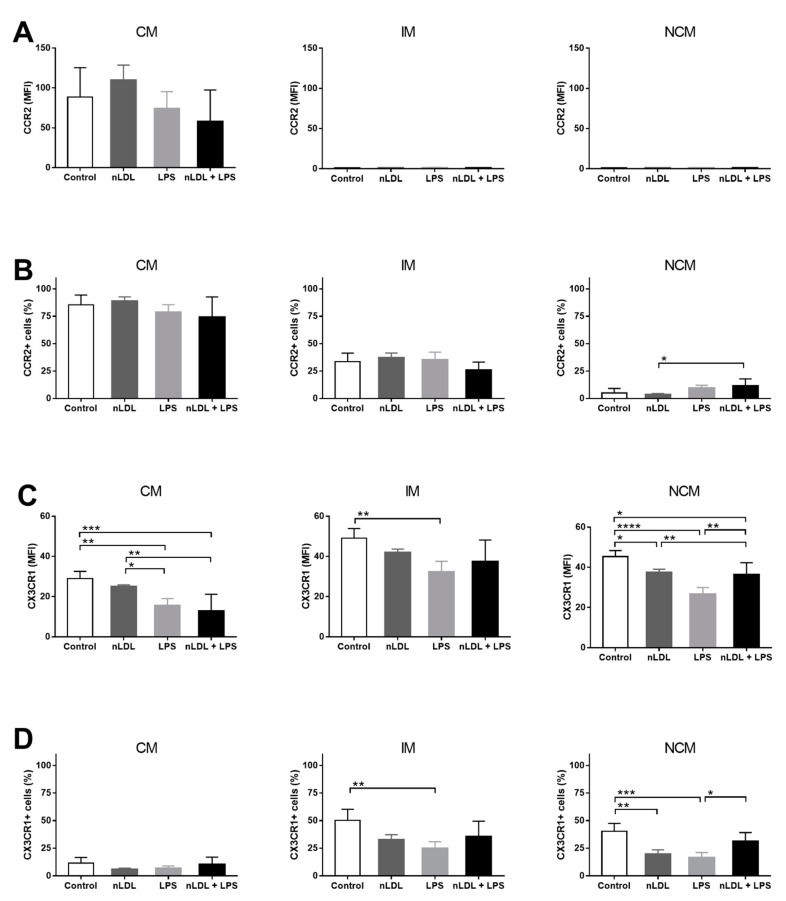
Effects of nLDL and LPS on CCR2 and CX3CR1 expression in human monocyte subpopulations. Although no significant differences were found, CCR2 expression was clearly higher in CMs than in IMs and NCMs (**A**, left, middle, and right panels, respectively). The number of CCR2+ cells increased in the CM subset as compared to IM and NCM subpopulations (**B**, left, middle, and right panels, respectively). In the NCM subset, LPS acted in synergy with nLDL to increase the amount of CCR2+ cells as compared to that found in cells only treated with nLDL (**B**, right panel). Expression of CX3CR1 increased in IMs and NCMs as compared to CMs (**C**, left, middle, and right panels, respectively). In NCMs, nLDL significantly decreased CX3CR1 expression as compared to control cells (**C**, right panel). LPS, alone or in combination with nLDL, decreased CX3CR1 expression in CMs, IMs, and NCMs (**C**, left, middle, and right panels, respectively). In IMs, LPS decreased the percentage of CX3CR1+ cells as compared to control cells (**D**, middle panel). In NCMs, nLDL significantly decreased the amount of CX3CR1+ cells as compared to control cells (**D**, right panel). Cells were incubated in the presence or absence of 100 μg/mL nLDL and/or 10 ng/mL LPS for 9 h. Data are expressed as mean ± standard deviation. Data were compared using one-way ANOVA followed by the post-hoc Tukey test. Differences were considered significant when *p* < 0.05. * = *p* < 0.05; ** = *p* < 0.01; *** = *p* < 0.001; **** = *p* < 0.0001. CM, classical monocytes; IM, intermediate monocytes; NCM, non-classical monocytes; nLDL, native low-density lipoproteins; LPS, lipopolysaccharide; CCR2, C-C chemokine receptor type 2; CX3CR1, CX3C chemokine receptor 1; MFI, median fluorescence intensity.

**Figure 5 biomolecules-11-01169-f005:**
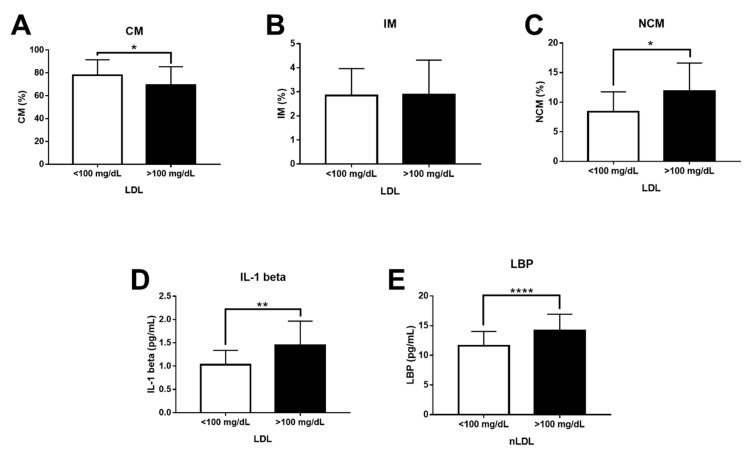
Levels of CMs, IMs, NCMs, IL-1 beta, and LBP in subjects with optimal and high serum levels of LDL-C. (**A**) The CM percentage significantly decreased in patients with serum LDL-C > 100 mg/dL (n = 85) as compared to controls (n = 65). (**B**) There were no significant changes between subjects with optimal and high LDL-C concentration for the IM percentage. (**C**) The NCM percentage significantly increased in patients with serum LDL-C > 100 mg/dL as compared to controls. (**D**) IL-1 beta serum levels significantly increased in patients with serum LDL-C > 100 mg/dL as compared to controls. (**E**) LBP serum levels significantly increased in patients with serum LDL-C > 100 mg/dL as compared to controls. Data are expressed as mean ± standard deviation. Data were compared using the unpaired Student’s *t*-test. Differences were considered significant when *p* < 0.05. * = *p* < 0.05; ** = *p* < 0.01; **** = *p* < 0.0001. CM, classical monocytes; IM, intermediate monocytes; NCM, non-classical monocytes; IL-1 beta, interleukin 1 beta; LBP, LPS-binding protein; LDL-C, low-density lipoproteins.

**Table 1 biomolecules-11-01169-t001:** Anthropometric and biochemical parameters of the study subjects. The study participants were divided into two groups of LDL-C concentration, as follows: subjects with optimal concentration of LDL-C ≤ 100 mg/dL, and individuals with high concentration of LDL-C > 100 mg/dL. Total cholesterol and Castelli’s atherogenic risk index significantly increased in patients with LDL-C greater than 100 mg/dL. Abbreviations: F, female; M, male; BMI, body mass index; LDL-C, low-density lipoproteins; HDL-C, high-density lipoproteins; HOMA-IR, homeostatic model assessment of insulin resistance; CRP, C-reactive protein; a.u., arbitrary units. Data are presented as mean ± standard deviation. Differences were considered significant when *p* < 0.05.

	LDL-C		
	<100 mg/dL	>100 mg/dL	*p*-Value
Sex (F/M)	28/37	41/44	0.2650
Age (years)	47.2 ± 1.056	49.49 ± 1.031	0.1913
BMI (kg/m^2^)	28.22 ± 1.172	28.05 ± 0.9035	0.9155
Waist circumference (cm)	91.58 ± 2.94	96.73 ± 2.002	0.1696
Body fat (%)	31.87 ± 1.696	31.91 ± 2.614	0.9909
Cholesterol (mg/dL)	166.7 ± 7.079	220.7 ± 4.723	<0.0001
Triglycerides (mg/dL)	242.6 ± 41.08	186.4 ± 11.14	0.0769
LDL-C (mg/dL)	76.53 ± 4.156	128.7 ± 3.653	<0.0001
HDL-C (mg/dL)	42.13 ± 3.44	47.06 ± 2.226	0.2332
Insulin (µU/L)	14.43 ± 1.057	13.52 ± 0.8046	0.5208
Glucose (mmol/L)	5.104 ± 0.2653	5.346 ± 0.2502	0.5675
HOMA-IR (a.u.)	3.299 ± 0.3261	3.126 ± 0.1866	0.6290
CRP (mg/L)	5.095 ± 0.4755	5.012 ± 0.6004	0.6328
Castelli’s risk index II	1.966 ± 0.1827	2.892 ± 0.1294	0.0002

## Data Availability

The data presented in this study are available upon request.

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
