# Peer review of "Native Low-Density Lipoproteins Act in Synergy with Lipopolysaccharide to Alter the Balance of Human Monocyte Subsets and Their Ability to Produce IL-1 Beta, CCR2, and CX3CR1 In Vitro and In Vivo: Implications in Atherogenesis"

_biomolecules, 2021, doi:10.3390/biom11081169_

Round 1

Reviewer 1 Report

Although the original question that the authors formulated and aimed to answer during the study could be relevant to the field, Manjarrez-Reyna et al. mainly perform three exvivo experiments where very few human whole blood cells markers are assessed: Il1B (Figure 3) and CCR2 and CXC3CR1 (Figure 3). The results obtained, are used to generate a complete hypothesis of the effect of nLDL and LPS on atherosclerosis.

The most positive aspect of the manuscript is that the authors used human cells. As expected, the cells somehow respond to these stimuli. However, the observations are not enough. The methodological approach is correct, but simple and not sufficient.

The study lacks ligand expression and mechanistic studies (i.e gene expression studies, migration assays, CCR2 and CXC3CR1 ligands, some signaling approaches…)

More comments, questions and suggestions:

  1. What is it known about the effect of nLDL on atherogenesis and recruitment of monocytes? and what is the relevance of LPS (bacterial infection/inflammatory settings) in the context of atherosclerosis? Several studies have been published already: i.e PMID: 23072373, PMID: 17548720 (some context is given in de disacusion section but a short explanation should also be added to the introduction.

58-63. Why do the authors focus on IL-1 beta, CCR2, and CX3CR1 and avoid/disregard other well-known inflammatory processes (p65 NFkB, TLR…) or migration pathways/markers (MIF1, ICAM…)??

  1. Materials and Methods section is very well written and very detailed.

Figure 2E. These comparisons between bar 3(>LPS) and bar 4 (nLDL+LPS) is always very similar, suggesting that the effect of nLDL+LPS is mainly, due to LPS itself. Please address this.

Figure 3 and Figure 4. These are good observations but the functional effect of these changes in the cell subtypes has not been addressed in this study. What about the ligand levels for CCR2, and CX3CR1???

Discussion: discussion should be toned down – this study does not address the migration properties of these cells. Marker expression might not be enough for such discussion.

Author Response

Reviewer #1

Although the original question that the authors formulated and aimed to answer during the study could be relevant to the field, Manjarrez-Reyna et al. mainly perform three exvivo experiments where very few human whole blood cells markers are assessed: Il1B (Figure 3) and CCR2 and CXC3CR1 (Figure 3). The results obtained, are used to generate a complete hypothesis of the effect of nLDL and LPS on atherosclerosis.

The most positive aspect of the manuscript is that the authors used human cells. As expected, the cells somehow respond to these stimuli. However, the observations are not enough. The methodological approach is correct, but simple and not sufficient.

The study lacks ligand expression and mechanistic studies (i.e gene expression studies, migration assays, CCR2 and CXC3CR1 ligands, some signaling approaches…)

Reply (R)

We thank to the Reviewer for her/his criticism. To date, the vast majority of studies have been focused on exploring the effect of oxLDL on the inflammatory ability of human macrophage-derived foam cells. However, evidence regarding the possible role of nLDL and LPS as key contributors to atherogenesis is still scarce. For this reason, the main goal of this first study was limited to examine the possible synergistic effect of nLDL and LPS on the expression of key inflammatory cytokines and chemokine receptors involved in atherogenesis such as IL-1 beta, CX3CR1, and CCR2. As the Reviewer correctly suggests, we are now performing additional studies aimed to characterize the possible molecular mechanisms through which nLDL and LPS can increase the inflammatory ability of human monocytes, contributing to the development of atherogenesis. However, results derived from the second part of experiments will be further reported.

Query (Q) 1

  1. What is it known about the effect of nLDL on atherogenesis and recruitment of monocytes? and what is the relevance of LPS (bacterial infection/inflammatory settings) in the context of atherosclerosis? Several studies have been published already: i.e PMID: 23072373, PMID: 17548720 (some context is given in de disacusion section but a short explanation should also be added to the introduction.

R1

To date, most of the studies regarding the role of LDL in the inflammatory response associated with atherogenesis has been focused on oxLDL and macrophages but not nLDL or monocyte subsets. However, accumulating evidence suggests that monocytes and nLDL could interact in circulation, thus contributing to the atherosclerotic plaque formation (Curr Pharm Des. 2018;24(26):3143-51, Sci Rep. 2016;6:20038). Likewise, LPS has been shown to enhance lipid accumulation in primary human adventitial fibroblasts and induce secretion of MCP-1, a chemokine with the ability to recruit monocytes toward atherosclerotic plaques; nevertheless, the possible synergistic role of LPS and nLDL in regulating the expression of inflammatory cytokines and chemokine receptors in human monocyte subpopulations remains uncertain.

Following the Reviewer’s observation, we added this contextual information in the Introduction section. Please find this information marked with yellow color at page 2.

Q2

58-63. Why do the authors focus on IL-1 beta, CCR2, and CX3CR1 and avoid/disregard other well-known inflammatory processes (p65 NFkB, TLR…) or migration pathways/markers (MIF1, ICAM…)??

R2

We decided to focus on IL-1 beta, CCR2, and CX3CR1 because these molecules (1) are well characterized in human monocyte subsets, (2) have been shown to play prominent roles in atherogenesis, and (3) are typical markers of inflammation and cell recruitment associated with human monocytes. However, the Reviewer’s observation is of great relevance, and it is worth mentioning that we are now working on measuring the levels of other proinflammatory cytokines and chemokine receptors such as TNF-alpha, IL-6, IL-17, and CCR5 as well as the signaling pathways orchestrating their effects. However, this is an ongoing work that we expect to report in a future communication.

Q3

  1. Materials and Methods section is very well written and very detailed.

R3

Thank you for your kind comments on this work.

Q4

Figure 2E. These comparisons between bar 3(>LPS) and bar 4 (nLDL+LPS) is always very similar, suggesting that the effect of nLDL+LPS is mainly, due to LPS itself. Please address this.

R4

Following the Reviewer’s suggestion, we added a sentence clarifying that the effect of LPS and nLDL, alone or in combination, on the percentage of CM, IM, and NCM is mainly attributed to LPS. Please find this information marked with red color at page 6. Thank you for your accurate observation.

Q5

Figure 3 and Figure 4. These are good observations but the functional effect of these changes in the cell subtypes has not been addressed in this study. What about the ligand levels for CCR2, and CX3CR1???

R5

We totally concur with the Reviewer. For this reason, we removed from the result section (with a special emphasis on Figures 3 and 4) all sentences and data interpretation that may extrapolate the results of IL-1 beta, CCR2, and CX3CR1 expression to functional effects. Please find these changes marked with green color at pages 8-10.

As mentioned above, we are now working on measuring the levels of several chemokines such as CCL2, CX3CL1, CCL13, and CCL8, among others. However, these experiments are still ongoing, and results will be reported in a future communication.

Q6

Discussion: discussion should be toned down – this study does not address the migration properties of these cells. Marker expression might not be enough for such discussion.

R6

Following the Reviewer’s observation, discussion has been toned down as suggested. Please find these changes marked with green color at pages 12 and 13.

We thank to the Reviewer for her/his very constructive questions and observations that have indubitably improved the last version of this manuscript.

Reviewer 2 Report

Thank you for your manuscript. 

Understanding how lipids impact circulating monocyte subsets’ atherogenic profile is important to discern given the high prevalence of dyslipidemia and CVD.  Thus, a study such as this is worth conducting. Overall, the manuscript is easy to read, however there are quite a few concerns as to the explanation and analysis of the data.

Abstract

  • The first line, ‘Accumulating evidence…. increase infiltration of IL-1 beta producing macrophages into atherosclerotic lesions’. This sentence is not strictly correct, as the cells come into the plaque from the circulation are monocytes that differentiate into macrophages. The issue can be solved by changing the word ‘infiltration’ to ‘accumulation’ or reword as appropriate correct the error. Actually, the first two lines in the discussion work well, and something similar could be used in the abstract.
  • I am not sure why the authors call LDL ‘native’ LDL, it doesn’t need to be nLDL, just LDL will suffice. When however, you are referring to LDL cholesterol measurement, it needs to be called LDL-C.

Introduction

The use of LPS: While LPS does stimulate monocytes and promote cytokine production, indeed it is the usual positive control in cytokine assays, what doesn’t come across in the introduction is why it is being tested in the study in combination with LDL.  So can the authors please provide some justification.  LPS is higher in the blood of patients, and is increased in the blood after a high fat meal, so add in some of references with this sort of information to support use of LPS.

Methods and results

  • The study was performed on 20 healthy donors whose LDL-C levels were optimal (<100mg/dL). However, these LDL-C levels are much larger than the 100ug/ml that their monocytes in the study were exposed to. Can the authors explain why 100ug/ml of LDL-C would be expected to have an effect on the monocytes? And how would these low values of LDL-C (to which the monocytes were exposed) have clinical relevance.
  • Why were the cells incubated for 9 hours; was a time course performed beforehand? Monocytes are normally treated with LPS for 4-6 hours. I appreciate that LDL would need longer, but it would be good to have a sentence that indicates whether this is the optimal time for LDL.
  • Where the donors fasting? As surface markers can vary depending on the fasted state.

2.2. Flow cytometer

The flow cytometry approach starts well, with whole blood approach used, blocking of non-specific binding, and use of comp beads. I have a few questions however about the flow plots and gating.  

  • The flow gating steps are not described in 2.2 Flow cytometry. Can you please add the steps in. Include how the gating quadrant positions for the subsets was decided. I appreciate that different studies vary in their gating approach, but the main thing is to be consistent within the one study, and I can’t see that here. There does not seem to be a set x or y axis value as they are different in the different plots in figure 2. While the expression levels of CD14 and CD16 may vary between people, and CD16 clearly does with culture, these quadrants should be fixed for the different culture conditions for the one person. In figure 2, the quadrant line is closer to the classical monocytes in B than it Is in A. As this inconsistency would alter the proportion reported, then the data in 2E is unreliable.  (Standard gating approaches are in the literature).
  • Furthermore, why are the CD14/CD16 plots shown in dot plots instead of pseudocolour -as had been used in the gating steps in figure 1, and is common in publications.
  • Can you please describe how marker percent positivity was determined? What was the cut off value for the control cells in the positive region, 0.1% or 2%?  
  • It is quite appropriate to report the percent positive cells for IL-1beta, as there are clearly two populations for all the culture conditions except for LPS. Can you clarify, when the MFI is reported for IL-1, is this of the positive population? Or have you included the negative population and reported it for all cells as a whole?
  • It is not appropriate to report the percentage positive cells for the surface markers (figure 4). This is because the histograms shown (in figure 2) are unimodal not bimodal. Ie there is a shift of the whole histogram – both the left and right side of the histogram have shifted compared to the left- and right-hand side of the control. There is not a clear negative and positive peak. Unless the left-hand side of the maker peak lines up with the left-hand side of the control peak, then it is unlikely that any of the cells are negative, rather any overlap of the marker histogram with the control peak indicates low  marker expression, weak antibody colour brightness or perhaps non optimal antibody binding.  As such 4b, and 4D are not relevant - at least not relevant for the histograms that are shown.
  • In figure 2, please show representative graphs for each subset. These will help the reader see the degree of expression for each subset. I feel this is important for the reader to see because the MFI that are reported in figure 4 are unusually small.
  • Can you please describe in the method how the MFI was obtained (that is used in figure 3 and 4). Was it the median or geometric mean? And, considering the small MFI, was it the MFI of the shifted peak alone, or was is the value reflective of taking away/ or dividing by a control peak.

The sentence (line 123) that mentions that for intracellular cytokine stain, WBCs were treated with Brefeldin A for the last 2 hours in culture, should be moved up to the section above. 2.1 In vitro culture of human primary monocytes.

Figure 5. The percentages reported will also be affected (as mentioned for figure 2) by whether there was consistent gating of the subset. So the reliability of this data may also be in question.

Discussion

The discussion contains a lot of information that belongs in an introduction, not in a discussion, as the information is not raised in discussion of the results.  For eg, we are introduced for to what LPS is, but it is not clear what result this information is explaining.   So please reassess what needs to go in the introduction and what should be in the discussion.

It is mentioned that LDL can alter the balance of monocyte populations directly by decreasing CM and increasing NCM, in vitro and in vivo, If this is the case, did the authors see a correlation between subset number or proportion and participant LDL levels?

Line 340, the authors say that nLDL might increase cD16 expression, but expression was not measured, subset proportions were. So please adjust this sentence to reflect the results.

Line 377 ‘the number of CCR2+NCM increased in response to nLDL and LPS suggesting increased ability of the cells to migrate into inflamed tissue’.   An increased number of cells, does not suggest they have an increased ability to migrate,  it would be an increased expression of the marker that would increase their ability. So please adjust the sentence to match the measure with its outcome.  Similarly address the same issue in the next sentence regarding increasing amount of CX3CR1 +NCM.  There is not a lot of discussion as to how these findings fit with the literature.  Others have looked at surface markers / cytokines in controls or patients relative to lipid levels. Please discuss your results in light of what has been found by others.

Author Response

Reviewer #2

Thank you for your manuscript. Understanding how lipids impact circulating monocyte subsets’ atherogenic profile is important to discern given the high prevalence of dyslipidemia and CVD.  Thus, a study such as this is worth conducting. Overall, the manuscript is easy to read, however there are quite a few concerns as to the explanation and analysis of the data.

Reply (R)

We thank to the Reviewer for her/his very kind comments regarding this manuscript.

Query (Q) 1

Abstract.     The first line, ‘Accumulating evidence…. increase infiltration of IL-1 beta producing macrophages into atherosclerotic lesions’. This sentence is not strictly correct, as the cells come into the plaque from the circulation are monocytes that differentiate into macrophages. The issue can be solved by changing the word ‘infiltration’ to ‘accumulation’ or reword as appropriate correct the error. Actually, the first two lines in the discussion work well, and something similar could be used in the abstract.

R1

Following the Reviewer’s suggestion, we replaced the word “infiltration” by “accumulation”. Please find this change marked with yellow color at page 1.

Q2

I am not sure why the authors call LDL ‘native’ LDL, it doesn’t need to be nLDL, just LDL will suffice. When however, you are referring to LDL cholesterol measurement, it needs to be called LDL-C.

R2

Thank you for your clever observation. Based on previous evidence, we refer to native LDL (nLDL) as the fraction of LDL that is not oxidized. In fact, numerous studies use this nomenclature to make the difference between oxidized LDL (oxLDL) and nLDL (Hypertension. 2007;50(2):276-83, doi: 10.1161/HYPERTENSIONAHA.107.089854; Sci Rep. 2018;8(1):11954, doi: 10.1038/s41598-018-30073-w; Hum. Immunol. 2010;71(8):737–744, doi: 10.1016/j.humimm.2010.05.005; Antioxid Redox Signal. 2010;13(1):39–75, doi: 10.1089/ars.2009.2733; among others).

In parallel and following the Reviewer’s suggestion, we replaced the term “LDL” by “LDL-C” when showing serum levels of low-density lipoproteins. Please find these changes marked with blue color at pages 1, 2, 4, and 10-12.

Q3

Introduction.       The use of LPS: While LPS does stimulate monocytes and promote cytokine production, indeed it is the usual positive control in cytokine assays, what doesn’t come across in the introduction is why it is being tested in the study in combination with LDL.  So can the authors please provide some justification.  LPS is higher in the blood of patients, and is increased in the blood after a high fat meal, so add in some of references with this sort of information to support use of LPS.

R3

Following the Reviewer’s suggestion, we added more references to support the use of LPS in this study. Please find this information marked with yellow color at page 2.

 Q4

Methods and results

The study was performed on 20 healthy donors whose LDL-C levels were optimal (<100mg/dL). However, these LDL-C levels are much larger than the 100ug/ml that their monocytes in the study were exposed to. Can the authors explain why 100ug/ml of LDL-C would be expected to have an effect on the monocytes?

R4

We thank to the Reviewer for her/his accurate observation. The use of 100 mg/ml nLDL and its in vitro effects on monocytes and non-immune cells have been consistently reported by numerous research teams (Sci Rep. 2016. 29;6:20038. doi: 10.1038/srep20038; BMC Med Genomics. 2008. 28;1:60. doi: 10.1186/1755-8794-1-60; Atherosclerosis. 2008;196(2):624-32. doi: 10.1016/j.atherosclerosis.2007.06.024). In fact, it has been reported that 100 mg/ml nLDL is able to increase cytokine expression, phagocytosis, and cell migration in monocytes. We speculate that even though 100 mg/ml nLDL is much less than that found in circulation as LDL-C, it is enough to be taken up via the LDL receptor (LDLr) and induce its effects on monocytes. Moreover, in this in vitro system the PCSK9 activity in charge of decreasing the amount of LDLr in the cell membrane is absent. Furthermore, in in vitro cultures we do not have the clearing system mediated by HDL through which huge LDL-C concentrations are constantly decreased, favoring that cells such as monocytes might be exposed to less LDL-C concentrations.

Q5

And how would these low values of LDL-C (to which the monocytes were exposed) have clinical relevance.

R5

We respectfully want to clarify that the main goal of this study was to test whether nLDL was able to affect human monocyte subsets in a direct fashion by altering the expression pattern of cytokine and chemokine receptors in vitro without necessarily attributing clinical relevance to the nLDL dose that we used. In fact, our findings show that in vitro exposure of human monocytes to 100 mg/ml nLDL increases the non-classical monocyte (NCM) percentage and decreases CX3CR1 expression and the classical monocyte (CM) percentage, which may help understand the role of CM, NCM and CX3CR1 in the inflammatory response associated with atherosclerotic lesion formation, a notion that increases the clinical relevance of this study. Please find this information marked with yellow color at page 12.

Q6

Why were the cells incubated for 9 hours; was a time course performed beforehand? Monocytes are normally treated with LPS for 4-6 hours. I appreciate that LDL would need longer, but it would be good to have a sentence that indicates whether this is the optimal time for LDL.

R6

Exposure time of in vitro cultures was selected based on time-response curves at 3, 6, and 9 hours, finding that monocytes show the most significant changes at 9 hours. Following the Reviewer’s recommendation, we added this information in the Material and Method section. Please find this information marked with green color at page 3.

Q7

Where the donors fasting? As surface markers can vary depending on the fasted state.

R7

Yes, all volunteers were enrolled in the study if they had 8-hours fasting. Following the Reviewer’s observation, we added this information in the Material and Method section. Please find this information marked with gray color at page 4.

Q8

2.2. Flow cytometer

The flow cytometry approach starts well, with whole blood approach used, blocking of non-specific binding, and use of comp beads. I have a few questions however about the flow plots and gating.

Q8a

The flow gating steps are not described in 2.2 Flow cytometry. Can you please add the steps in.

R8a

Following the Reviewer’s suggestion, we added the flow gating steps in the 2.2 section of Materials and Methods. Please find this information marked with pink color at page 3.

Q8b

Include how the gating quadrant positions for the subsets was decided. I appreciate that different studies vary in their gating approach, but the main thing is to be consistent within the one study, and I can’t see that here. There does not seem to be a set x or y axis value as they are different in the different plots in figure 2. While the expression levels of CD14 and CD16 may vary between people, and CD16 clearly does with culture, these quadrants should be fixed for the different culture conditions for the one person. In figure 2, the quadrant line is closer to the classical monocytes in B than it Is in A. As this inconsistency would alter the proportion reported, then the data in 2E is unreliable.  (Standard gating approaches are in the literature).

R8b

The gating quadrant position was performed according to previous reports (A. M. Zawada et al., “Comparison of Two Different Strategies for Human Monocyte Subsets Gating Within the Large-Scale Prospective CARE FOR HOMe Study,” Cytom. Part A, vol. 70, 2015, doi:10.1002/cyto.a.22703), wherein each monocyte subpopulation can be recognized using the rectangular gating strategy. Following the Reviewer’s suggestion, we added this information in the Material and Method section. Please find this information marked with pink color at page 3.

Furthermore, due to a terrible mistake we submitted preliminary dot plots containing inaccurate gating quadrant positions in Figure 2. Thanks to your extraordinary observation, we have detected this mistake and now, we show the final dot plots that served for performing all flow cytometry analyses. As you will be able to see, the gating quadrant positions did not vary among the study subjects. Please find these changes in new Figure 2 at page 6.

Q8c

Furthermore, why are the CD14/CD16 plots shown in dot plots instead of pseudocolour -as had been used in the gating steps in figure 1, and is common in publications.

R8c

Following the Reviewer’s suggestion, we replaced dot plots by pseudocolor plots. Please find this change in new Figure 2 at page 6.

Q8d

Can you please describe how marker percent positivity was determined? What was the cut off value for the control cells in the positive region, 0.1% or 2%? 

R8d

Thank you for this important question. The cut-off value for cells in the positive region was 0.1%, according to FMO control tubes.

Q8e

It is quite appropriate to report the percent positive cells for IL-1beta, as there are clearly two populations for all the culture conditions except for LPS. Can you clarify, when the MFI is reported for IL-1, is this of the positive population? Or have you included the negative population and reported it for all cells as a whole?

R8e

We reported MFI for IL-1 beta from both positive and negative cell populations (as a whole). Following the Reviewer’s observation, we added this information in the Material and Method section. Please find this information marked with turquoise color at page 3.

Q8f

It is not appropriate to report the percentage positive cells for the surface markers (figure 4). This is because the histograms shown (in figure 2) are unimodal not bimodal. Ie there is a shift of the whole histogram – both the left and right side of the histogram have shifted compared to the left- and right-hand side of the control. There is not a clear negative and positive peak. Unless the left-hand side of the maker peak lines up with the left-hand side of the control peak, then it is unlikely that any of the cells are negative, rather any overlap of the marker histogram with the control peak indicates low  marker expression, weak antibody colour brightness or perhaps non optimal antibody binding.  As such 4b, and 4D are not relevant - at least not relevant for the histograms that are shown.

R8f

We respectfully want to clarify that reporting the percentage of positive cells from unimodal histograms is commonly performed and accepted, especially for surface markers of monocyte subpopulations such as chemokine receptors (please see Geng S et al., Nat Commun. 2016;8(7):13436; Filipovic I et al., Front Immunol. 2019;19(10):2692; Mukherjee R et al., Sci Rep. 2015;5:13886; Devevre EF et al., J Immunol. 2015;194(8):3917-23; and Chamorro S et al., MAbs. 2014;6(4):1000-12, among others). This can be explained by the fact that increase in positive cells is quantified as the proportion of displacement from the origin to the right side along the x axis, according to FMO controls and not culture well controls. For this reason, we decided to leave panels 4B and 4D that in turn are accompanied by panels 4A and 4C, wherein we also show MFI for CCR2 and CX3CR1.

Q8g

In figure 2, please show representative graphs for each subset. These will help the reader see the degree of expression for each subset. I feel this is important for the reader to see because the MFI that are reported in figure 4 are unusually small.

R8g

Following the Reviewer’s suggestion, we added representative graphs for each monocyte subset. Please find this change in new Figure 2 at page 6.

Q8h

Can you please describe in the method how the MFI was obtained (that is used in figure 3 and 4). Was it the median or geometric mean? And, considering the small MFI, was it the MFI of the shifted peak alone, or was is the value reflective of taking away/ or dividing by a control peak.

R8h

We obtained MFI values as the median fluorescence intensities in both positive and negative cell populations (as a whole), without dividing by control peak. Following your suggestion, we added this information in the Material and Method section. Please find this information marked with turquoise color at page 3.

Q8i

The sentence (line 123) that mentions that for intracellular cytokine stain, WBCs were treated with Brefeldin A for the last 2 hours in culture, should be moved up to the section above. 2.1 In vitro culture of human primary monocytes.

R8i

Following the Reviewer’s suggestion, we moved up the abovementioned sentence to the section 2.1 of the Material and Method section. Please find this information marked with yellow color at page 3.

Q9

Figure 5. The percentages reported will also be affected (as mentioned for figure 2) by whether there was consistent gating of the subset. So the reliability of this data may also be in question.

R9

As we mentioned above, due to a terrible mistake we submitted preliminary dot plots containing inaccurate gating quadrant positions in Figure 2. However, your observation allowed us detecting this mistake and now, we show the final dot plots that served for performing all flow cytometry analyses. It is worth mentioning that the gating quadrant positions did not vary among the study subjects of Figures 2 and 5. Thank you again for your opportune criticism.

Q10

Discussion. The discussion contains a lot of information that belongs in an introduction, not in a discussion, as the information is not raised in discussion of the results.  For eg, we are introduced for to what LPS is, but it is not clear what result this information is explaining.   So please reassess what needs to go in the introduction and what should be in the discussion.

R10

Following the Reviewer’s observation, we made contextual changes in both the Introduction and Discussion sections, having special emphasis on providing evidence on the possible role of LPS and nLDL in atherogenesis. Please find this information marked with yellow color at page 2, 12, and 13.

Q11

It is mentioned that LDL can alter the balance of monocyte populations directly by decreasing CM and increasing NCM, in vitro and in vivo, If this is the case, did the authors see a correlation between subset number or proportion and participant LDL levels?

R11

We performed Pearson’s correlation analyses and found no significant associations of LDL-C serum levels with percentages of CM (r=-0.2, P=0.08), IM (r=-0.03, P=0.40), and NCM (r=0.09, P=0.25). For this reason, we originally decided not to include this information in the manuscript.

Q12

Line 340, the authors say that nLDL might increase cD16 expression, but expression was not measured, subset proportions were. So please adjust this sentence to reflect the results.

R12

Following the Reviewer’s suggestion, we rephrased the abovementioned sentence to avoid data misinterpretation. Please find this information marked with yellow color at page 12.

Q13

Line 377 ‘the number of CCR2+NCM increased in response to nLDL and LPS suggesting increased ability of the cells to migrate into inflamed tissue’.   An increased number of cells, does not suggest they have an increased ability to migrate,  it would be an increased expression of the marker that would increase their ability. So please adjust the sentence to match the measure with its outcome.

R13

We totally concur with the Reviewer. For this reason, we removed not only from the Discussion section but also the Result section all sentences that may extrapolate the results of CCR2 and CX3CR1 expression to the migratory ability of monocytes. Please find these changes marked with green color at pages 8, 9, 10, 12, and 13.

Q14

Similarly address the same issue in the next sentence regarding increasing amount of CX3CR1 +NCM.  There is not a lot of discussion as to how these findings fit with the literature.  Others have looked at surface markers / cytokines in controls or patients relative to lipid levels. Please discuss your results in light of what has been found by others.

R14

Following the Reviewer’s suggestion, we have discussed and compared our own findings in light of previous evidence. Please find this information marked with green color at pages 12 and 13.

We sincerely thank to the Reviewer for her/his criticism and very constructive comments on this work. We believe that your observations, comments, and suggestions have indubitably improved the last version of the manuscript.

Reviewer 3 Report

Manjarrez-Reyna and colleagues have explored the synergy between nLDL and LPS in in vitro and in vivo settings.

The hypothesis is interesting and in line with recent observations indicating that nLDL may have differential vascular effects depending on the presence of additional factors (Rheumatology (Oxford). 2021 Feb 1;60(2):866-871).

Major comment:

  1. In the in vitro culture of primary human monocytes, nLDL is obtained from 20 healthy donors. As lipoprotein particles can carry LPS themselves, it would be advisable to measure LDL-LPS content or, at least, have some biochemical data on the donors relative to their inflammatory state. For example, ultra-sensitive CRP.
  2. Quite similarly, in relation to the 150 volunteers of the in vivo study, although those with anti-inflammatory or immunomodulatory medication are excluded, a inflammatory marker to ensure that both groups are comparable would be advisable.

I understand that all volunteers are healthy but, still the information is necessary as it is central to the hypothesis of the study. If this information is not available, this aspect should be addressed in the discussion.

3. Another potentially important factor is LDL particle number and size. In discordant subjects with similar LDL concentrations these parameters might be different and it is well demonstrated their effect on atherosclerosis. This can be assessed by measuring apoB. If that is not possible, please add a comment to the discussion. 

Author Response

Reviewer #3

Manjarrez-Reyna and colleagues have explored the synergy between nLDL and LPS in in vitro and in vivo settings. The hypothesis is interesting and in line with recent observations indicating that nLDL may have differential vascular effects depending on the presence of additional factors (Rheumatology (Oxford). 2021 Feb 1;60(2):866-871).

Reply (R)

We thank you for your kind comments on this work.

Query (Q) 1

In the in vitro culture of primary human monocytes, nLDL is obtained from 20 healthy donors. As lipoprotein particles can carry LPS themselves, it would be advisable to measure LDL-LPS content or, at least, have some biochemical data on the donors relative to their inflammatory state. For example, ultra-sensitive CRP.

R1

We respectfully want to clarify that nLDL for in vitro culture of primary monocytes was not obtained from healthy donors. In fact, nLDL was purchased from Sigma-Aldrich (Sigma Aldrich, St. Louis, MO, USA) in order to avoid contamination with other lipoproteins or LPS, as occurs when using whole blood-isolated LDL. Please see this information marked with red color at page 3. However, we believe the Reviewer’s comment is of great relevance and we added values of high-sensitivity C-reactive protein of healthy donors (n=20) that provided monocytes for in vitro cultures. In fact, healthy donors showed high-sensitive C-reactive protein serum values of 1.35±0.26 mg/l, on average. Please fin this information marked with green color at page 2.

Q2

Quite similarly, in relation to the 150 volunteers of the in vivo study, although those with anti-inflammatory or immunomodulatory medication are excluded, a inflammatory marker to ensure that both groups are comparable would be advisable. I understand that all volunteers are healthy but, still the information is necessary as it is central to the hypothesis of the study. If this information is not available, this aspect should be addressed in the discussion.

R2

We already measured CRP serum levels in all in vivo study subjects to guarantee they had no active infection or inflammatory disease but decided not to include this information in the original submission. However, we think the Reviewer’s comments is of great importance and we decided to include this information in the revised version of the manuscript. In fact, we found no significant differences between controls and subjects with elevated LDL for CRP serum levels (5.095 ± 0.4755 mg/l versus 5.012 ± 0.6004 mg/l, P=0.6328, respectively). Please find this information marked with dark yellow color at pages 4, 5, and 10, and in Table 1.

Q3

3. Another potentially important factor is LDL particle number and size. In discordant subjects with similar LDL concentrations these parameters might be different and it is well demonstrated their effect on atherosclerosis. This can be assessed by measuring apoB. If that is not possible, please add a comment to the discussion.

R3

Thank you for your important observation. We concur that the LDL particle number and size is a matter of great relevance; unfortunately, we did not measure apoB levels in this study. For this reason and following the Reviewer’s suggestion, we added a sentence to the Discussion section pointing out this pending work. Please find this information marked with red color at page 12.

We thank you for your very constructive comments on this work. Your criticism has indubitably improved the last version of the manuscript.

Round 2

Reviewer 1 Report

No more comments. 

Reviewer 2 Report

Thank you, 

My concerns were thoroughly addressed. 

Reviewer 3 Report

All questions have been adequately addressed.